# Molecular Characterization of *Salmonella* spp. Isolates from Wild Colombian Babilla (*Caiman crocodilus fuscus*) Isolated In Situ

**DOI:** 10.3390/ani12233359

**Published:** 2022-11-30

**Authors:** Roy Rodríguez-Hernández, María Paula Herrera-Sánchez, Julián David Ortiz-Muñoz, Cristina Mora-Rivera, Iang Schroniltgen Rondón-Barragán

**Affiliations:** 1Poultry Research Group, Faculty of Veterinary Medicine, University of Tolima, Altos the Santa Helena, A.A 546, Ibagué 730006299, Colombia; 2Immunobiology and Pathogenesis Research Group, Faculty of Veterinary Medicine, University of Tolima, Altos the Santa Helena, A.A 546, Ibagué 730006299, Colombia; 3Biodiversity and Dynamics of Tropical Ecosystems Research Group, University of Tolima, Altos the Santa Helena, A.A 546, Ibagué 730006299, Colombia

**Keywords:** caiman, in situ, resistance genes, *Salmonella*, virulence genes

## Abstract

**Simple Summary:**

*Salmonella* spp. is a major foodborne pathogen responsible for salmonellosis in animals and humans. The gastrointestinal tracts of reptiles and amphibians have been described as a source of *Salmonella* transmission, and the main outbreaks that have been reported occurred after contact with infected animals as pets or after the ingestion of their meat. The aim of this study was to determine the presence of 24 virulence genes and characterize the genotypic antibiotic resistance profile in *Salmonella* strains isolated from *Caiman crocodilus fuscus* obtained in situ (natural habitat) in Prado, Tolima, (Colombia) in a previous study. Fifteen *Salmonella* strains were obtained from *Caiman crocodilus fuscus*, where all the strains have 3 out of 26 resistances genes and 19 out of 24 virulence genes. The results indicate that Colombian babilla (*Caiman crocodilus fuscus*) may have a role as a carrier of multidrug resistant bacteria.

**Abstract:**

*Salmonella* enterica is a pathogen capable of colonizing various environments, including the intestinal tract of different animals such as mammals, birds, and reptiles, which can act as carriers. *S. enterica* infection induces different clinical diseases, gastroenteritis being the most common, which in some cases, can evolve to septicemia and meningitis. Reptiles and amphibians have been reported as a reservoir of *Salmonella*, and transmission of the pathogen to humans has been documented. This study aimed to determine the presence of virulence genes and characterize the genotypic antibiotic resistance profile in *Salmonella* strains isolated from *Caiman crocodilus fuscus* obtained in situ (natural habitat) in Prado, Tolima, Colombia in a previous study and stored in a strain bank in our laboratory. Fifteen *Salmonella* strains were evaluated through endpoint PCR to determine the presence of resistance genes and virulence genes. The genes *bla*_TEM_, *strB*, and *sul1* were detected in all the strains that confer resistance to ampicillin, streptomycin, and sulfamethoxazole, as well as the virulence genes *invA*, *pefA*, *prgH*, *spaN*, *tolC*, *sipB*, *sitC*, *pagC*, *msgA*, *spiA*, *sopB*, *sifA*, *lpfA*, *csgA*, *hilA*, *orgA*, *iroN*, *avrA*, and *sivH*, indicating the possible role of babilla (*Caiman crocodilus fuscus*) as a carrier of multidrug-resistant bacteria.

## 1. Introduction

The most global biodiversity is concentrated in seventeen countries referred to as megadiverse, and Colombia is one of the richest countries regarding biological diversity worldwide [1]. However, human activities causing landscape-scale loss, disruption to natural habitats, and pollution generate changes in biological resources, increasing interactions between humans and wildlife. A great variety of reptile species carrying pathogens of interest to human health have gained popularity as pets, becoming possible vectors of antibiotic-resistant bacteria in the environment [2,3,4].

*Salmonella enterica* is a pathogen capable of colonizing various environments, including the intestinal tract of different animals, such as mammals, birds, and reptiles including lizards, snakes, and chelonians, which can be bacteria’s carriers for human contagion [5]. Currently, three species are recognized for the genus *Salmonella*: *S. bongori*, *S. enterica*, and *S. subterranean*. They are subdivided into subspecies and serotypes [6,7]. Regarding *S. bongori*, 22 serotypes associated with cold-blooded animals have been found, and in some cases, these infect humans and cause disease [8,9]. Nowadays, 2700 serotypes of *S. enterica* have been described, *S. enterica* subsp. *enterica* being the most frequently isolated from warm-blooded and cold-blooded animals. It is associated with 99% of *Salmonella* infections in humans and animals [10,11].

*S. enterica* infection induces different clinical diseases depending on the host immune response [12]. Gastroenteritis is the most common disease caused mainly by Typhimurium and Enteritidis serotypes, which, in some cases, can evolve into septicemia and meningitis, mainly in immunocompromised patients, pregnant women, and elderly adults [13]. In addition, the symptoms in humans vary depending on the host immunity and virulence factors of the strain that play a crucial role in systemic infections due to the activation during the interaction between the bacteria and the hostile environment of the gastrointestinal tract of the host [14]. It is estimated that *Salmonella* causes 1.35 million cases of illness and 420 deaths in the United States per year [15]. Vegetable-type foods and food of animal origins, such as pig, chicken, cow, and lamb meat, and eggs or milk, are considered the main source of *Salmonella* infections in humans, since production animals are in contact with wild animals, insects, or contaminated food, allowing the disease to be classified as a disease of zoonotic origin [16,17,18]. 

*Salmonella* is naturally found in the gastrointestinal tract of reptiles and amphibians; these animals have been described as a source of *Salmonella* transmission, and the main outbreaks that have been reported occurred after direct or indirect contact with animals kept as pets or after the ingestion of their meat [19]. In Colombia, it has been reported that the contamination of *Salmonella* spp. in reptiles, such as *Caiman crocodilus*, is due to the fact that bacteria colonize them systemically via water, soil, contaminated food, and/or through vertical transmission as its entry route [20]. The *Caiman crocodilus* is a reptile that is found in the wild and was domesticated for production due to its economic value for its meat and skin [21]. Some *Salmonella* serotypes frequently associated with reptiles and isolated in humans are *S.* Poona, *S.* Weltevreden, *S.* Enteritidis, *S.* Java, *S.* Stanley, and subspecies *S. enterica diarizonae*, all known to have caused infection in humans and produced diarrhea in most cases [19,22]. 

*Salmonella* is considered a public health concern due to different animals that carry the bacteria and have a function as vehicles of transmission, as well as its resistance to multiple antibiotics (MDR) that hinders its treatment [4,23]. Previously, the resistance genes *aadA* (aminoglycosides), *sul2* (sulphonamides), *dfr1*, *dfr7* (trimethoprim), and *tet*(A) and *tet*(B) (tetracyclines), which confer resistance to antibiotics used in humans, have been reported in *Salmonella* isolated from different species of reptiles [24]. In waterbodies where wildlife animals live, it has been revealed the antibiotic pollution accumulated in the sediments caused by wastewater from animal farming and hospitals is a problem due to the development of bacteria’s resistance to multiple antibiotics and horizontal gene transfer [25]. Therefore, research is necessary to obtain a better understanding of the genotypic profile of the *Salmonella* spp. isolates in order to establish effective prevention and control strategies. The aim of this study was to determine the presence of virulence genes and characterize the genotypic antibiotic resistance profile of *Salmonella* strains isolated from Colombian babilla (*Caiman crocodilus fuscus*) in situ, located in Prado, Tolima, (Colombia).

## 2. Materials and Methods

### 2.1. Study Population and Sample Collection

The Colombian babilla (*Caiman crocodilus fuscus)* has a natural habitat in the Hidroprado hydroelectric dam of the Prado, Tolima region of Colombia, which is located between the central and eastern mountains of the Colombian Andes. Fifteen (*n* = 15) *Salmonella* strains, stored in our laboratory strain bank and isolated from cloacal swabs of wild Colombian babilla collected in a previous study in our laboratory, were used for characterizing the genotypic antibiotic resistance and determination of virulence genes in this study [26,27]. The strains used were serotyped in previous research as *Salmonella* Paratyphi B (*n* = 2), *Salmonella* Saintpaul (*n* = 2), *Salmonella* Javiana (*n* = 4), *Salmonella* Braenderup (*n* = 4), *Salmonella* Soerenga (*n* = 1), *Salmonella* Infantis (*n* = 1), and *Salmonella* Powell (*n* = 1) [26]. No ethical approval was required for this study because *Salmonella* spp. strains were from the Bacterial Strain Collection of the Laboratory of Immunology and Molecular Biology of the Universidad del Tolima.

### 2.2. Genomic DNA Extraction and Molecular Confirmation of Salmonella Isolates

Genomic DNA (gDNA) was extracted from fresh bacterial colonies using Invisorb^®^ Spin Universal Kit (Stratec, Berlin, Germany). The isolated gDNA was stored at −20 °C until its analysis. All strains were confirmed as *Salmonella* by amplification of a 284 bp fragment of the *invA* gene (accession number: M90846.1) by endpoint PCR, using *S*. Enteritidis (ATCC 13076^®^) as positive control and *E. coli* (ATCC 25922^®^) as negative control (Table 1). 

### 2.3. Determination of Virulence Genes and Genotypic Antibiotic Resistance

Genes related to virulence factors (Table 1) and antibiotic resistance (Table 2) were screened by endpoint PCR. In the case of the virulence genes, the *S*. Enteritidis (ATCC 13076^®^) was used as a positive control and in the resistance genes, the positive controls correspond to *Salmonella* spp. strains previously characterized in the laboratory of immunology and molecular biology. The reactions were carried out in a ProFlex PCR System (Applied Biosystems, Carlsbad, CA, USA) using 25 μL of total reaction volume, composed by 1 μL of gDNA, 5 μL of Flexi Buffer 5x colorless GoTaq ^®^ (Promega, Madison, WI, USA), 1 μL of dNTPs (Invitrogen, Carlsbad, CA, USA), 1 μL of each primer (forward and reverse) (Table 1 and Table 2) (10 pmol/Μl) (Macrogen, Seoul, Korea), 1 μL of MgCl_2_ (25 mM) (Promega, Madison, WI, USA), 0.125 μL of GoTaq Flexi DNA Polymerase (Promega, Madison, WI, USA), and 14,875 μL of nuclease-free water. The amplification conditions were an initial denaturation at 95 °C for 3 min, followed by 35 cycles of 30 s of denaturation at 95 °C, annealing step at 55 °C for 30 s, an extension step at 72 °C for 30 s, and a final extension step of 7 min at 72 °C. The annealing temperature and the extension step time were adjusted according to the primer set and length of each amplicon. The amplification products were revealed on 2% agarose gel electrophoresis (PowerPac™ HC, Bio-Rad, Hercules, CA, USA), using 100 bp DNA Ladder (NEB, Ipswich, MA, USA) and HydraGreen™ (ACTGene, Piscataway, NJ, USA) as the intercalating agent. Finally, the gels were visualized and documented using the ENDURO™ GDS gel documentation system (Labnet International, Edison, NJ, USA).

## 3. Results

### 3.1. Distribution of Virulence Genes among Salmonella

All 15 *Salmonella* isolates from wild Colombian babilla (*Caiman crocodilus fuscus*) were positive for *invA* (Figure 1), *pefA, prgH, spaN, tolC, sipB, sitC, pagC, msgA, spiA, sopB, sifA, lpfA, csgA, hilA, orgA, iroN, avrA*, and *sivH* virulence-associated genes (Table 3). Likewise, *lpfC* (*n* = 13; 86.7%) and *cdtB* (*n* = 11; 73.3%) genes were found in more than 73% of the *Salmonella* isolates. Additionally, *sopE* (*n* = 5; 33.3%) and *spvB* (*n* = 1; 6.67%) genes were found, but at a lower frequency. Furthermore, none of the isolates was positive for the *sefA* gene, and overall, the 15 *Salmonella* isolates were positive for at least ten virulence genes (Table 3). 

### 3.2. Genotypic Antibiotic Resistance in Salmonella Isolates

The genes *bla*_TEM_, *strB*, and *sul1* that confer resistance to ampicillin, streptomycin, and sulfamethoxazole, respectively, were amplified in all the strains (Table 4). Likewise, the genes conferring resistance to ceftriaxone were found in high frequency (*bla*_CMY2_: 86.7%; *bla*_CTX-M_: 80%), as was the *sul2* gene (*n* = 12; 80%), which confers resistance to sulfamethoxazole. Furthermore, the genes *dfrA1* (*n* = 4; 26.7%), *floR* (*n* = 1; 6.67%), and *qnrD* (*n* = 1; 6.67%) were found in low frequency. Moreover, none of the genes that encode for gentamicin, quinolones, and fluoroquinolones resistance were detected in the isolates. 

## 4. Discussion

*Salmonella* has been isolated from different animals, such as impala, sable, leopard, and reptiles, including alligators, snakes, lizards, and turtles in captivity and as pets [4,32,33]. This bacterium is naturally found in the gastrointestinal tract of animals, and the outbreaks that have been reported occurred after direct or indirect contact with animals as pets or as meat [34]. Previously, Zając et al. [33] reported the presence of *Salmonella* in different reptile species (85.5%; *n* = 597/696) and pet reptiles, such as snakes (76.3%; *n* = 16/21), lizards (69%; *n* = 33/48), and chelonians (19%; *n* = 10/54) [4]. In Colombia, the serotypes of *Salmonella* reported by López-Cruz et al. [26] and obtained from Colombian babilla were similar to those reported in other animals, such as broiler chickens, known to be responsible for the disease in humans [30,35,36]. It has been reported that feeding with contaminated chicks and rodents is a factor that promotes the spread of *Salmonella* common serotypes. This fact could be related to this study because the animals were located in a dam near farms with domestic animals that, in some cases, become food for the babillas [33]. Furthermore, *S*. Paratyphi B and *S*. Braenderup have been reported in reptiles such as snakes and turtles [37]. Particularly, *S.* Paratyphi B is a serotype of special concern due to its ability to infect humans and induce paratyphoid fever [37,38]. In addition, *S*. Saintpaul has been isolated from fruits such as mangoes and vegetables such as tomatoes, which play a role in several outbreaks in humans [39,40].

In the genotypic resistance, the *bla*_TEM_ gene was found in all the isolates (*n* = 15/15). The *bla*_TEM_ gene encodes for Extended Spectrum Beta-lactamase (ESBL), a protein that produces a hydrolyzation of β-lactam antibiotics such as ampicillin [41]. This frequency is higher compared to the *Salmonella* strains isolated from iguanas, where two out of eight carried the gene [42]. In addition, Bittner-Torrejón [43] reported that 61.7% (*n* = 21/34) of the strains isolated from reptiles were positive for *bla*_TEM_ gene. Previously, Marin et al. [4] showed that the phenotypic resistance to ampicillin (46.7%; *n* = 35/75) was the third most common resistance in pet reptiles that could be transmitted due to direct or indirect contact between humans and pets. The babillas used in this study were found in the wild, where contact with humans is not common. However, the antibiotic pollution is an increasing problem that generates MDR bacteria and depends on the antibiotics used in the community; in the case of Colombia, ampicillin, an antibiotic of the penicillin group, is a common antibiotic prescribed to patients, classified in the access group of antibiotics [44]. 

On the other hand, *strB*, a gene that encodes for enzymes that inactivate streptomycin [45] was found in all the isolates (*n* = 15/15), which differs from other studies where this gene was reported in *Salmonella* associated with poultry, swine, cattle, horses, wild reptiles, wild mammals, and companion animals but with a lower frequency (34.7%; *n* = 67/193) [46]. Moreover, in Chile, no reptiles presented the gene [43]. In reptiles of Poland, *Salmonella* strains present a frequent resistance to streptomycin (25%; *n* = 134/533) [33] as well as reptiles in Lithuania (26%; *n* = 13/50) [24]. Nevertheless, in Lithuania the phenotypic resistance is mediated by other genes, such as *armA* and *aadA*, that were not evaluated in this study [24]. Likewise, the *sul1* gene encoding dihydropteroate synthases, which confer sulfamethoxazole resistance [47], was detected in the totality of the strains, which differs from reports in the United States, where this gene was not present in wild reptiles [46]. Furthermore, *bla*_CMY2_ (86.7%; *n* = 13/15) and *bla*_CTX-M_ (80%; *n* = 12/15) genes that confer resistance to ceftriaxone and cefotaxime through the hydrolyzation of antibiotics have been reported in *Salmonella* isolated from iguanas (*bla*_CTX-M_*; n* = 1/8), horses, wild reptiles, wild mammals, and companion animals (*bla*_CMY2_*; n* = 44/193) [42,46]. 

In our study, all 15 *Salmonella* strains present virulence genes related to the host invasion, host recognition, and colonization, such as *invA*, *pefA*, *prgH*, *spaN*, *tolC*, *sipB*, *sitC*, *pagC*, *msgA*, *spiA*, *sopB*, *sifA*, *lpfA*, *csgA*, *hilA*, *orgA*, *iroN*, *avrA*, *and sivH* (*n* = 19/24), which are identified as responsible for the pathogenesis of the bacteria during salmonellosis [48]. Dudek et al. [49] reported fifteen virulence genes in *Salmonella* strains (*invA*, *prgH*, *orgA*, *tolC*, *sitC*, *spiA*, *spiB*, *spaN*, *iroN*, *IpfC*, *sifA*, *sopB*, *pagC*, *cdtB* and *msgA*) isolated from reptiles (*n* = 15/84). Additionally, in Poland, *Salmonella* from wild birds (*n* = 64/1000) was reported to contain *spiA*, *msgA*, *invA*, *lpfC*, and *sifA* genes [50]. However, even though the detection of virulence genes is used as a tool to predict the virulence of the bacteria, the presence of the genes does not necessarily confer greater pathogenicity because in order to increase it, the combined expression of several genes is required [51,52]. 

## 5. Conclusions

The presence of several resistance genes in *Salmonella* strains isolated from Colombian babilla (*Caiman crocodilus fuscus)* could describe this reptile as a carrier of multidrug-resistant bacteria that has genotypic resistance genes, such as *bla*_TEM_, *strB*, and *sul1* genes; likewise, *bla*_CMY2_ and *bla*_CTX-M_ genes were found in 80% of the isolates. In addition, *Salmonella* isolates from wild babilla are carriers of a wide number of virulence genes and are necessary to develop more investigations of wild reptiles on this topic. 

## Figures and Tables

**Figure 1 animals-12-03359-f001:**
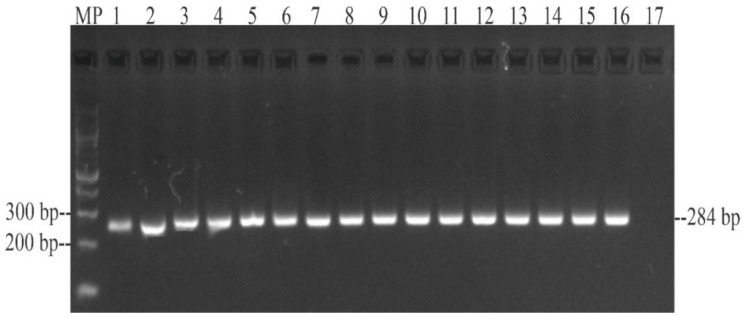
PCR amplification of a 284 bp fragment from the *invA* gene of *Salmonella* isolated from wild *Caiman crocodilus fuscus*. MP—100 bp DNA ladder (New England Biolabs, Ipswich, MA, USA); lane 1: *S.* Paratyphi B; lane 2: *S.* Paratyphi B; lane 3: *S.* Saintpaul; lane 4: *S.* Saintpaul; lane 5: *S.* Javiana; lane 6: *S.* Javiana; lane 7: *S.* Javiana; lane 8: *S.* Javiana; lane 9: *S.* Braenderup; lane 10: *S.* Braenderup; lane 11: *S.* Braenderup; lane 12: *S.* Braenderup; lane 13: *S.* Soerenga; lane 14: *S.* Infantis; lane 15: *S.* Powell; lane 16: positive control *S.* Enteritidis ATCC 13076^TM^; 17: negative control *E. coli* ATCC 25922^TM^.

**Table 1 animals-12-03359-t001:** Primer sequences used for amplification of virulence genes in *Salmonella* spp. isolates.

Virulence Factor	Gene	Primer Sequence		Amplicon Size (bp)	References
Structure (The invasion-associated type III secretion system)	*invA*	F	GTGAAATTATCGCCACGTTCGGGCAA	284	Webber et al. [14]
		R	TCATCGCACCGTCAAAGGAACC		
	*prgH*	F	GCCCGAGCAGCCTGAGAAGTTAGAAA	756	
		R	TGAAATGAGCGCCCCTTGAGCCAGTC		
	*spaN*	F	AAAAGCCGTGGAATCCGTTAGTGAAGT	504	
		R	CAGCGCTGGGGATTACCGTTTTG		
	*sipB*	F	GGACGCCGCCCGGGAAAAACTCTC	875	
		R	ACACTCCCGTCGCCGCCTTCACAA		
Effector protein (The invasion-associated type III secretion system)	*sopB*	F	CGGACCGGCCAGCAACAAAACAAGAAGAAG	220	
		R	TAGTGATGCCCGTTATGCGTGAGTGTATT		
	*sifA*	F	TTTGCCGAACGCGCCCCCACACG	449	
		R	GTTGCCTTTTCTTGCGCTTTCCACCCATCT		
Regulatory protein (The invasion-associated type III secretion system)	*hilA*	F	CTGCCGCAGTGTTAAGGATA	497	
		R	CTGTCGCCTTAATCGCATGT		
Fimbriae	*lpfC*	F	GCCCCGCCTGAAGCCTGTGTTGC	641	
		R	AGGTCGCCGCTGTTTGAGGTTGGATA		
	*pefA*	F	GCGCCGCTCAGCCGAACCAG	157	
		R	GCAGCAGAAGCCCAGGAAACAGTG		
	*lpfA*	F	CTTTCGCTGCTGAATCTGGT	250	
		R	CAGTGTTAACAGAAACCAGT		
	*csgA*	F	TCCACAATGGGGCGGCGGCG	350	
		R	CCTGACGCACCATTACGCTG		
	*sefA*	F	GATACTGCTGAACGTAGAAGG	488	
		R	GCGTAAATCAGCATCTGCAGTAGC		
Plasmid	*spvB*	F	CTATCAGCCCCGCACGGAGAGCAGTTTTTA	717	
		R	GGAGGAGGCGGTGGCGGTGGCATCATA		
Survival inside cells	*tolC*	F	TACCCAGGCGCAAAAAGAGGCTATC	161	
		R	CCGCGTTATCCAGGTTGTTGC		
	*pagC*	F	CGCCTTTTCCGTGGGGTATGC	454	
		R	GAAGCCGTTTATTTTTGTAGAGGAGATGTT		
	*msgA*	F	GCCAGGCGCACGCGAAATCATCC	189	
		R	GCGACCAGCCACATATCAGCCTCTTCAAAC		
	*spiA*	F	CCAGGGGTCGTTAGTGTATTGCGTGAGATG	550	
		R	CGCGTAACAAAGAACCCGTAGTGATGGATT		
Toxins	*cdtB*	F	ACAACTGTCGCATCTCGCCCCGTCATT	268	
		R	CAATTTGCGTGGGTTCTGTAGGTGCGAGT		
Iron metabolism	*sitC*	F	CAGTATATGCTCAACGCGATGTGGGTCTCC	768	
		R	CGGGGCGAAAATAAAGGCTGTGATGAAC		
	*iroN*	F	ACTGGCACGGCTCGCTGTCGCTCTAT	1205	Skyberg et al. [28]
		R	CGCTTTACCGCCGTTCTGCCACTGC		
Structure (The invasion-associated type III secretion system)	*orgA*	F	TTTTTGGCAATGCATCAGGGAACA	255	
		R	GGCGAAAGCGGGGACGGTATT		
Effector protein (The invasion-associated type III secretion system)	*avrA*	F	AGCCTGGCGCTCGCCAAAAA	123	This study
		R	GCGGTCTGCTTTATCGGACGGG		
	*sopE*	F	GAGGGCCGGGCAGTGTTGAC	121	
		R	CTTCACGGGTCTGGCTGGCG		
	*sivH*	F	AGCGCGCTGAATGCGGTGAT	121	
		R	TCTTGTCGCGCCACAGCAGG		

**Table 2 animals-12-03359-t002:** Primer sequences used for amplification of resistance genes in *Salmonella* spp. isolates.

Antibiotic	Gene	Primer Sequence		Amplicon Size (bp)	References
Ampicillin	*bla* _PSE-1_	F	GCAAGTAGGGCAGGCAATCA	422	Chuanchuen and Padungtod [29]
		R	GAGCTAGATAGATGCTCACAA		
	*bla* _TEM_	F	ATCAGTTGGGTGCACGAGTG	608	
		R	ACGCTCACCGGCTCCAGA		
Chloramphenicol	*catA*	F	CCAGACCGTTCAGCTGGATA	454	
		R	CATCAGCACCTTGTCGCCT		
	*cmlA*	F	TGGACCGCTATCGGACCG	641	
		R	CGCAAGACACTTGGGCTGC		
Florfenicol	*floR*	F	CACGTTGAGCCTCTATATGG	888	
		R	ATGCAGAAGTAGAACGCGAC		
Gentamicin	*aadB*	F	CTAGCTGCGGCAGATGAGC	300	
		R	CTCAGCCGCCTCTGGGCA		
Streptomycin	*aadA1*	F	CTCCGCAGTGGATGGCGG	631	
		R	GATCTGCGCGCGAGGCCA		
	*aadA2*	F	CATTGAGCGCCATCTGGAAT	500	
		R	ACATTTCGCTCATCGCCGGC		
	*strA*	F	TGGCAGGAGGAACAGGAGG	405	
		R	AGGTCGATCAGACCCGTGC		
	*strB*	F	GCGGACACCTTTTCCAGCCT	621	
		R	TCCGCCATCTGTGCAATGCG		
Trimethoprim	*dfrA1*	F	CAATGGCTGTTGGTTGGAC	254	
		R	CCGGCTCGATGTCTATTGT		
	*dfrA10*	F	TCAAGGCAAATTACCTTGGC	432	
		R	ATCTATTGGATCACCTACCC		
	*dfrA12*	F	TTCGCAGACTCACTGAGGG	330	
		R	CGGTTGAGACAAGCTCGAAT		
Tetracycline	*tet*(A)	F	GCTGTCGGATCGTTTCGG	658	
		R	CATTCCGAGCATGAGTGCC		
Sulfamethoxazole	*sul1*	F	CGGACGCGAGGCCTGTATC	591	
		R	GGGTGCGGACGTAGTCAGC		
	*sul2*	F	GCGCAGGCGCGTAAGCTGAT	514	
		R	CGAAGCGCAGCCGCAATTC		
	*sul3*	F	GGGAGCCGCTTCCAGTAAT	500	
		R	TCCGTGACACTGCAATCATTA		
Ceftriaxone	*bla* _CMY2_	F	AAATCGTTATGCTGCGCTCT	224	Castro-Vargas et al. [30]
		R	CCGATCCTAGCTCAAACAGC		
	*bla* _CTX-M_	F	TTCGCTAAATACCGCCATTC	236	
		R	TATCGTTGGTTGTGCCGTAA		
Quinolones and fluoroquinolones	*oqxA*	F	GGTGAAGTCGATCAGTCAGT	154	
		R	ATCTATCGTGAACAGCACCT		
	*qnrA*	F	CCGCTTTTATCAGTGTGACT	188	
		R	ACTCTATGCCAAAGCAGTTG		
	*qnrB*	F	GATCGTGAAAGCCAGAAAGG	469	Herrera-Sánchez et al. [31]
		R	ACGATGCCTGGTAGTTGTCC		
	*qnrC*	F	GGGTTGTACATTTATTGAATCG	308	
		R	CACCTACCCATTTATTTTCA		
	*qnrD*	F	CGAGATCAATTTACGGGGAATA	582	
		R	AACAAGCTGAAGCGCCTG		
	*qnrS*	F	ACGACATTCGTCAACTGCAA	417	
		R	TAAATTGGCACCCTGTAGGC		
	*aac(6′)-Ib*	F	TTGCGATGCTCTATGAGTGGCTA	482	
		R	CTCGAATGCCTGGCGTGTTT		

**Table 3 animals-12-03359-t003:** Virulence genes profiles of *Salmonella* spp. isolates.

Virulence Factor	Gene	*Salmonella* Serotypes Positive for Virulence Genes															Total (%)
		*S.* Paratyphi B	*S.* Paratyphi B	*S.* Saintpaul	*S.* Saintpaul	*S.* Javiana	*S.* Javiana	*S.* Javiana	*S.* Javiana	*S.* Braenderup	*S.* Braenderup	*S.* Braenderup	*S.* Braenderup	*S.* Soerenga	*S.* Infantis	*S.* Powell	
Structure	*prgH*																15 (100)
	*spaN*																15 (100)
	*sipB*																15 (100)
	*orgA*																15 (100)
Effector protein	*sopB*																15 (100)
	*sifA*																15 (100)
	*avrA*																15 (100)
	*sopE*																5 (33.3)
	*sivH*																15 (100)
Regulatory protein	*hilA*																15 (100)
Fimbriae	*lpfC*																13 (86.7)
	*pefA*																15 (100)
	*lpfA*																15 (100)
	*csgA*																15 (100)
	*sefA*																0 (0)
Plasmid	*spvB*																1 (6.67)
Survival inside cells	*tolC*																15 (100)
	*pagC*																15 (100)
	*msgA*																15 (100)
	*spiA*																15 (100)
Toxins	*cdtB*																11 (73.3)
Iron metabolism	*sitC*																15 (100)
	*iroN*																15 (100)

Black box = gene is present; box = gene is absent.

**Table 4 animals-12-03359-t004:** Genotypic antibiotic resistance profiles of *Salmonella* spp. isolates. Black box = gene is present; box = gene is absent.

Antibiotic	Gene	*Salmonella* Serotypes Positive for Resistance Genes															Total (%)
		*S.* Paratyphi B	*S.* Paratyphi B	*S.* Saintpaul	*S.* Saintpaul	*S.* Javiana	*S.* Javiana	*S.* Javiana	*S.* Javiana	*S.* Braenderup	*S.* Braenderup	*S.* Braenderup	*S.* Braenderup	*S.* Soerenga	*S.* Infantis	*S.* Powell	
Ampicillin	*bla* _PSE-1_																7 (46.7)
	*bla* _TEM_																15 (100)
Chloramphenicol	*catA*																0 (0)
	*cmlA*																9 (60)
Florfenicol	*floR*																1 (6.67)
Gentamicin	*aadB*																0 (0)
Streptomycin	*aadA1*																0 (0)
	*aadA2*																0 (0)
	*strA*																0 (0)
	*strB*																15 (100)
Trimethoprim	*dfrA1*																4 (26.7)
	*dfrA10*																0 (0)
	*dfrA12*																0 (0)
Tetracycline	*tet*(A)																7 (46.7)
Sulfamethoxazole	*sul1*																15 (100)
	*sul2*																12 (80)
	*sul3*																0 (0)
Ceftriaxone	*bla_CMY2_*																13 (86.7)
	*bla_CTX-M_*																12 (80)
Quinolones and fluoroquinolones	*oqxA*																0 (0)
	*qnrA*																0 (0)
	*qnrB*																0 (0)
	*qnrC*																0 (0)
	*qnrD*																1 (6.67)
	*qnrS*																0 (0)
	*aac(6′)-Ib*																0 (0)

## Data Availability

Not applicable.

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
