# Peer review of "Molecular Characterization of Salmonella spp. Isolates from Wild Colombian Babilla (Caiman crocodilus fuscus) Isolated In Situ"

_animals, 2022, doi:10.3390/ani12233359_

Round 1

Reviewer 1 Report

Abstract- lines 31-38 number of virulence genes and antibiotic resistance genes found in the study does not provide much value for the abstract. Authors should highlight important genes or other findings that provide information to the reader.  What is the prevalence, how many animals were screened, where they were located, and which important virulence and AMR genes were found?

Lines 41-48 This intro paragraph seems irrelevant. Salmonella is quite persistent and found in animals. This is not new information. Please

Lines 44-57 Again, there is no point to start from bongori while the entire manuscript focuses on enterica subsp. Instead, authors should spend their effort on providing background on Salmonella enterica in reptiles and their environment. Is the Colombian babilla a wild reptile or a pet? The introduction should provide enough background about the possible public health risks related to Salmonella enterica found in Colombian babilla.

Line 76- reference 20 is not appropriate as this citation refers to the molecular typing tool. Please find an appropriate citation obtained from global organizations. AMR in Salmonella is a global problem and authors (until now) used very country-specific (e.g., Denmark, Italy, Czech Republic) or non-appropriate citations to refer global Salmonella problem. I would revise the references here and cite the global papers where the problem is reflected worldwide. Otherwise, please refer to the problem from the country standpoint where the study was performed.

Line 77-78 – Authors did not provide any introduction to the prevalence of genes and AMR profiles observed in reptiles until here. The such intro should be provided. Also, why focus on genotypic profiles but not phenotypic ones? Authors should provide background work (if published by others in the same region or country) or nearby countries or the rest of the world.   As a reader, I couldn’t understand how prevalent Salmonella is in reptiles if it is big problem in Colombia, what AMR or virulence traits were observed in such animal species, and if there was any previous study conducted. Please revise the entire introduction.

Materials and methods: sampling frame should be justified. 80 animals were screened for Salmonella, how they were detected, caught, and selected? Where are they located? How were the cloacal swabs collected?

Line 88- the previous project citation is not appropriate. Please provide a source for the readers to understand the previous project on which this study was sourced. Also, this guideline is for food-chain, please provide your methodology.

Line 90- Only using the White-Kauffmann-Le Minor scheme? Any reagents? This is not the way to present the “Materials and methods” please clarify and be specific. The microbiology section of this manuscript requires major clarifications.

Line 94-95 is this an animal use protocol /approval? If so, please clarify.

Line 97- To me, the entire virulence gene selection seems unclear. Are these known important virulence genes that result in infection in humans?  This needs to be clarified. Also, why not whole-genome sequencing? I don’t see any “molecular confirmation of Salmonella isolates in this section”. Was there any negative control used for the PCR?

Table 1- Please group these genes by virulence mechanisms and add a column for mechanism.   The manuscript should justify why these genes were selected for screening.

Table 2- gene names should be revised. Please group these genes by antibiotic class and add a column for class.  The manuscript should justify why these genes were selected for screening.

Table 3- I am having difficulty understanding this table. For example. invA gene should be present in all Salmonella enterica spp. It is a gene used for the confirmation, similarly, all other genes that exist across all Salmonella serotypes can be preserved as genes that are found across all Salmonella. If that is the case, there is no point to show these genes. Also, the positive control for the sefA gene did not exist. How author ensure this PCR was working?

Table 4- Similar situation exists for Table 4, was there any positive and negative controls for these genes? Also, ParathypiB is not a non-typhoidal Salmonella author should clarify why they included ParathypiB in their study. Again, the gene names in this table and the rest of the manuscript should be revised.

Line 171- higher than?

Lines 197-200 these lines should be moved to the introduction and studies should be provided in the introduction as a background. It is not clear, which research gap this study aimed to fill.

Line 201-203 rather than talking about ampicillin, authors should talk about the AMR profiles observed in Salmonella and their relationship with the treatment of Salmonella in humans

Line 221- what is n number here?

Line 220-227 authors should be able to discuss the function of their finding (virulence genes) and also have some background about the genes that are conservative in the entire Salmonella population (e.g., invA) as such genes do not have any discriminative power.

Reviewer 2 Report

This is a really interesting and novel article about antimicrobial profile in babilla. Most AMR studies in wildlife are performed by disk diffusion or broth microdilution. Very few authors investigate the presence of resistance genes (ARGs), probably due to budget issues. The advantage of detecting resistance genes, apart from the immediacy of the results, is that it allows detection of  ARGs that may not be expressed and can be transferred to other bacteria in the environment. Therefore, this study becomes even more important.

However, before publishing it, the manuscript would have to be improved. One of the things that have shocked me the most is the statement in the abstract that reptiles are the main reservoir of Salmonella for humans, when in fact is poultry.

Line 51 indicates that "mammals, birds, reptiles, and insects are the main reservoirs"... I think this sentence is very pretentious. It is true that all homeothermic species are potential carriers of Salmonella, but they are not reservoirs. For an animal to be a reservoir, it must be a carrier and also be capable of transmitting it to humans, that is, there must be a direct or indirect interaction between that animal and humans. I suggest modifying that sentence.

Other suggestions are:

- L63: please add "and" before "food of animal originis".

- L100: please change "use" to "analysis".

- Table 1: it is of great interest to know the reference from which the first employees have been extracted. However, in the primers of avrA, sopE and sivH it indicates "This study". So, I understand that these primers have been specifically designed for the study and therefore have not been used before in any other study. Is it, right?

- The first time the common name of the Caiman crocodilus fuscus is mentioned is on line 130. I think it should be mentioned the first time this species is mentioned, and for the rest of the text please use the common name and not the Latin name.

- Tables 3 and 4. My congratulations. Both tables are very visual, which is not always easy to achieve. In addition, I greatly appreciate the detail of indicating the antibiotic in Table 4.

- L164: the bacterium does not live in the tract of "some" animals but in the tract of practically all homeothermic animals and even some ectotherms. Please reformulate the sentence.

- L166-187. This part of the discussion does not correspond to this article. Here data that are being discussed are not found in the material and methods, nor in the results. They correspond to a previous study. Please remove this. If you want to mention direct the reader to those studies with a reference.

- L202. Is it possible that a comma is missing after "penicillin group"?

- L221-223. I can't find out if n=15 refers to Dudek's study or yours. Please reformulate.

- L232: this study is not confirming the interaction between the babilla and the human being that allows the transmission of Salmonella and therefore the reservoir character of the babilla. Just that he's a carrier. In addition, again, the serotypes found in the study prior to this one are mentioned. The conclusions must allude to the data obtained in this study: pathogenicity factors and resistance genes. Please reformulate.

Round 2

Reviewer 1 Report

Line 52. “human contamination” is not appropriate. Please revise.

Line 77- “Salmonella sp.” Needs to chance to spp.

Line 92 “production of bacteria’s resistance” is not appropriate. please revise this sentence.

Line 95 “this bacterium” is not appropriate. It is spp. not a single bacterium.

I cannot locate lane 17 E. coli negative control in the Figure 1. Please revise. Also, this information (invA gene finding) was also provided in Table3. Please consider providing Figure 1 as a supporting information or consider to remove invA from table 3.  

Still few AMR genes (bla and tet genes) are not presented correctly in table 4 and in the manuscript. Please use subscripts and parentheses as appropriate. e.g., blaTEM   tet(A)

Lines 235-241 According to this comparison, Dudek et al. did not find invA gene in Salmonella is this correct? Are these studies looked for the same genes as the authors did? To make comparisons, the methodologies should match. Please revise.  

Reviewer 2 Report

The authors have considerably improved the manuscript, however there are still some small modifications to be made:

Separate the Simple Summary and Abstract into two paragraphs.

Throughout the text, the authors write about Caiman crocodillus fuscus and suddenly in line 154 the Colombian babilla appears. What is Colombian babilla? Is it the same animal? Please it is essential to relate the scientific and common name the first time you mention an animal species on line 96 as follows: Colombian Babilla (Caiman crocodillus fuscus) and from that point use only the common name. In line 76, change "Caiman crocodillus" to "babilla (Caiman crocodillus)", and in line 78, replace the scientific name with the common one.

L. 163. Figure 1 does not appear.

L. 192-207. I don't quite understand this... When I read the manuscript for the time, I supposed that the strains used in this study were isolated and characterized in a study prior to this one (reference 26). The article under review corresponds to the AMR and virulence analyses, not to the isolation and serotyping. If so, why include and discuss results already published in a final degree project? Please remove these lines from the discussion. It makes no sense to discuss something that has already been discussed previously in another publication...

L. 194 mentions Marin et al. (reference 4) as a study that includes crocodile samples when it is not true. It includes chelonians, lizards, and snakes, but not crocodilians. Please remove that reference from here.

L. 240. "Dudek et al. [51] reported only six virulence genes"

L. 241-242. The article focuses on reptiles, specifically on babillas. It makes no sense now to talk about the genes found in birds... Please delete this sentence.

L. 251-253. This conclusion on serotypes is again from the study previously published in the final degree project. It does not apply to this study.
